# Abnormal Social Interactions in a *Drosophila* Mutant of an Autism Candidate Gene: *Neuroligin 3*

**DOI:** 10.3390/ijms21134601

**Published:** 2020-06-29

**Authors:** Ryley T. Yost, J. Wesley Robinson, Carling M. Baxter, Andrew M. Scott, Liam P. Brown, M. Sol Aletta, Ramtin Hakimjavadi, Asad Lone, Robert C. Cumming, Reuven Dukas, Brian Mozer, Anne F. Simon

**Affiliations:** 1Department of Biology, Faculty of Science, Western University, London, ON N6A 5B7, Canada; ryost@uwo.ca (R.T.Y.); jrobi8@uwo.ca (J.W.R.); lbrow55@uwo.ca (L.P.B.); maletta@uwo.ca (M.S.A.); rhakimja@uwo.ca (R.H.); mlone@uwo.ca (A.L.); rcummin5@uwo.ca (R.C.C.); 2Animal Behaviour Group, Department of Psychology, Neuroscience and Behaviour (PNB) McMaster University, Hamilton, ON L8S 4K1, Canada; baxtercm@mcmaster.ca (C.M.B.); scottam3@mcmaster.ca (A.M.S.); dukas@mcmaster.ca (R.D.); 3Office of Research Integrity, Office of the Assistant Secretary for Health, Rockville, MD 20889, USA; Brian.Mozer@hhs.gov

**Keywords:** *neuroligin*, sex, social behavior, social space, sociability, aggression, climbing, *Drosophila* Stress Odorant (dSO) avoidance

## Abstract

Social interactions are typically impaired in neuropsychiatric disorders such as autism, for which the genetic underpinnings are very complex. Social interactions can be modeled by analysis of behaviors, including social spacing, sociability, and aggression, in simpler organisms such as *Drosophila melanogaster*. Here, we examined the effects of mutants of the autism-related gene *neuroligin 3* (*nlg3*) on fly social and non-social behaviors. Startled-induced negative geotaxis is affected by a loss of function *nlg3* mutation. Social space and aggression are also altered in a sex- and social-experience-specific manner in *nlg3* mutant flies. In light of the conserved roles that neuroligins play in social behavior, our results offer insight into the regulation of social behavior in other organisms, including humans.

## 1. Introduction

Social interactions amongst individuals within a group are important for the development of normal social behavior, which can have strong effects on survival and reproduction. An individual’s ability to respond to others relies on the perception of social cues and their subsequent integration in the nervous system [1]. We are only beginning to understand how the neural circuitry, or the brain’s ability to process signals from others, facilitates this integration and allows for a response to other individuals [2]. The inability to properly integrate social cues can result in abnormal social interactions, as seen in individuals with autism spectrum disorders [3,4]. However, the genetic and molecular mechanisms involved in the integration of cues and responses to other individuals remain poorly understood.

Since it can be difficult to study social behaviors in animals such as humans and other mammals due to their complexity, *Drosophila* is an excellent model to study the underlying mechanisms driving social behavior [5]. Although less complex than mammals, *Drosophila* exhibit behaviors that are shared by more complex organisms, such as learning and memory, sleep, and aggression, to name a few [5,6,7]. These conserved behaviors make *Drosophila* a good model to study how cues from other individuals are integrated within the brain. In addition, *Drosophila* can be used to study environmental stimuli that alter interactions with others [7,8].

Recently, social space has been used as an effective assay to study social interactions with others in a group to identify neural circuitry, underlying mechanisms, and functional changes within the fly brain that modulate these interactions [2]. Other social behaviors, including courtship, aggression, sociability, and *Drosophila* Stress Odorant (dSO) avoidance, are also used to examine the underlying mechanisms of social behavior in *Drosophila* [9,10,11,12,13].

Studies have focused on identifying the sensory modalities, neural circuitry, and synaptic components that alter social space. Social space requires vision but not classical olfaction [14]. Neural circuitry, such as specific dopaminergic clusters of neurons, contribute to social space [15]. At the synaptic level, multiple pre- and post-synaptic proteins modulate social space. Mutants of *tyrosine hydroxylase* (*th*), required for dopamine synthesis, *catecholamines up (catsup)*, and *vesicular monoamine transporter* (*vmat*) alter social space [16]. In the post-synaptic neuron, expression of *rugose (rg)*, a homolog of the human gene *neurobeachin*, an autism candidate gene, is important for social space, as well as expression of *forkhead box P* (*foxP)*, and *fragile X-related 1 (fmr1),* a homolog of the Fragile X mental retardation gene [17,18,19]. Finally, the *neuroligins* (*nlg*), have also been shown to affect social space [20]. Although several genetic modifiers of social space have been identified, the environment also plays a major role in regulating social behavior. Exposure to environmental toxins, such as BPA [21]; increased age of parents and progeny [22]; and previous social experiences, including social isolation, increases social space [14].

The *neuroligin (nlg)* gene family encodes postsynaptic cell adhesion proteins that regulate the development, maturation, and function of the synapse [23,24]. In *Drosophila*, there are four paralogs of *nlg*. While *nlg1* encodes a protein localized to the larval neuromuscular junction (NMJ)*, nlg2, nlg3,* and *nlg4* encode proteins located in the central nervous system (CNS) neurons and NMJ [25,26,27,28]. Based on modENCODE data from Flybase, *nlg2, nlg3,* and *nlg4,* are expressed beginning at the embryo stage and continues to be expressed in the adult head until at least 20 days old (maximum age reported) [29,30]. The *nlgs* are found in different synapse types. Both *nlg1* and *nlg4* are found predominantly in excitatory synapses, while *nlg2* is expressed in inhibitory synapses. Unlike the other *Drosophila* paralogs, *nlg3* is found in both excitatory and inhibitory synapses, making *nlg3* unique and of interest. In the CNS, Nlg4 localizes to clock neurons [31]; however, Nlg2- and Nlg3- specific localization has yet to be determined. Mutations within murine *nlg* genes result in impairments in social interactions and communication and increased repetitive behavior, key characteristics of autism spectrum disorders [32]. *Drosophila* social behavior is also affected by mutations within *nlg2* and *nlg4* genes. While *nlg4* knockout flies display decreased social space, both *nlg2* and *nlg4* mutants prefer smaller and larger group sizes, respectively [20]. Little is known about the behavioral consequences of a loss of *nlg3* expression in *Drosophila*. Larval and adult locomotion decreases in *nlg3* knockout flies [28,33]. Nlg3 is cleaved within the extracellular acetylcholinesterase-like domain by the protease Tumor necrosis factor α-converting enzyme (TACE) resulting in two protein variants, Nlg3-Full length (FL) and Nlg3-Short (S) [33]. Interestingly this cleavage is neuron specific and does not occur at the NMJ [33]. The only known functional difference between the protein variants is that the short variant is required for proper locomotor activity and operates in a neurexin-independent manner [33]. However, the role of *nlg3* in *Drosophila* social behavior remains unknown.

A previous microarray analysis study revealed that *nlg3* was the only *Drosophila* paralog to exhibit expression changes with prior social experience; male flies exhibited increased *nlg3* transcript abundance after a period of social isolation [referred to as CG34127 in [34]. Since *nlg2* and *nlg4* mutants alter the response to others in social space [20], and *nlg3* transcript abundance might increase after isolation, we wondered whether *nlg3* influences social space in response to environmental changes.

In this study, we investigated the role of *nlg3* in *Drosophila* social behavior using several assays to assess social interactions in groups of flies: social space [35] and sociability [11]. We compared these assays to the response of flies in other social contexts, including aggression [36] and dSO avoidance [37]. Finally, we examined the role of *nlg3* in the modulation of social space in response to moderate aging and social isolation.

## 2. Results

### 2.1. Nlg3 Protein Abundance Is Not Altered in Mutants or with Age

Western blot analysis revealed Nlg3 protein abundance in three *nlg3* mutants: a deficiency line (thereafter *nlg3^Def1^*), a line with a P-element insertion into the fourth intron (*nlg3^L04^*), and a line with P-element insertion into the regulatory region of the gene (*nlg3^GS32^*–Figure 1A). As previously reported [28], we also detected two bands, representing full-length protein (Nlg3-FL) and a short isoform of the protein (Nlg3-S), arising from cleavage of the full-length protein after translation (Figure 1A,B) [33]. We assessed Nlg3 levels in mutants at two different ages (3–4 days old compared to 7–10 days old), in both sexes (Figure 1B–E). As expected, the loss-of-function *nlg3^Def1^* line displayed no detectable Nlg3 protein (Figure 1A,B).

In males, there was significantly less Nlg3-S than Nlg3-FL in our control line Canton-S (Cs) (the genetic background in which all mutants were outcrossed, Figure 1D; effect of protein isoform: *p* = 0.0072). However, we found no effect of moderate aging on either Nlg3 isoforms in Cs flies. In contrast, the opposite effect was observed for the male mutants *nlg3^L04^* and *nlg3^GS32^*: the difference in quantity between Nlg3 isoforms was absent in young males, but at 7–10 days old both isoforms decreased in amount (effect of age: *p* = 0.0344).

In Cs females, there was no difference between Nlg3-FL and Nlg3-S (*p* = 0.7403), no significant effect of age (*p* = 0.0727), and finally, no expression differences within the mutants (Figure 1E; *p* = 0.3587).

### 2.2. nlg3 Affects Social Space in a Sexually Dimorphic Manner with Age

As social space is affected in *nlg2* and *nlg4* mutants [20], we first tested social space in *nlg3* mutants. Male *nlg3^Def1^* exhibited increased social space, as demonstrated by fewer flies within four body lengths (4BL; ~1 cm) regardless of age (Figure 2A; *p* = 0.0227). However, there was no effect of *nlg3^Def1^* in 3–4 day-old females, and a reduction of flies within 4BL was seen only at 7–10 days old (Figure 2B; *p* = 0.0276). In contrast, regardless of age or sex, there was no significant effect of either *nlg3^L04^* or *nlg3^GS32^* on social spacing (Figure 2C–F).

Finally, when a *nlg3* cDNA was overexpressed in all neurons using a pan-neuronal driver (UAS-*nlg3*/+; *elav*-Gal4/+; thereafter denoted as *elav* > *nlg3*), both males and females were affected in an age-specific manner. In young males, both UAS-*nlg3/+* and *elav* > *nlg3* caused a decrease in the number of flies within 4BL (Figure 2G; effect of genotype: *p* = 0.0024). Older *elav* > *nlg3* males were not different from the controls (interaction of age and genotype: *p* = 0.0045). Young *elav* > *nlg3* females were not different from their controls (Figure 2H; *p* = 0.0644), but when older, an increase in flies within 4BL in UAS-*nlg3*/+ and *elav* > *nlg3* was observed (effect of age: *p* = 0.0047; interaction between age and genotype: *p* = 0.0379). 

To confirm that *elav* > *nlg3* indeed leads to an overexpression of Nlg3, we performed Western blot analysis on the drivers and overexpression line. The UAS-*nlg3*/+ line itself exhibited elevated Nlg3 expression at twice the amount of the control, displaying a leaky effect. However, Nlg3 expression in the *elav* > *nlg3* line was ~6 times higher than the control (Appendix A).

Since there was no effect on social space in *nlg3^L04^* or *nlg3^GS32^* mutants, we focused the rest of our analysis on *nlg3^Def1^* mutants, in addition to flies overexpressing *nlg3*.

### 2.3. nlg3 Alters Startle-Induced and Spontaneous Locomotor Activity

Previously, a loss of function of *nlg3* has been shown to reduce locomotor activity [28,33]. Hence, we wanted to confirm previous findings and validate our loss-of-function mutant. Using a climbing assay to investigate startle-induced activity, we observed age-related reductions in climbing in Cs, as well as in UAS-*nlg3/+* flies, as expected [2], in both males and females (Figure 3A–D).

In contrast, male *nlg3^Def1^* flies had reduced climbing at a young age compared to Cs (Figure 3A; effect of genotype: *p* = 0.0428). However, no difference in climbing ability between Cs and *nlg3^Def1^* male flies was observed at 7–10 days old (effect of age: *p* = 0.0045; interaction between age and genotype: *p* = 0.0365). Reduced climbing was seen in female *nlg3^Def1^* flies, at both ages (Figure 3B; effect of genotype: *p* = 0.0075; effect of age: *p* < 0.0001). 

Pan-neuronally overexpressing *nlg3* did not affect climbing ability in young males (Figure 3C); however, older *elav* > *nlg3* males did not show an effect of age (Figure 3C; effect of age: *p* < 0.0001; interaction between age and genotype: *p* = 0.0267). Females *elav* > *nlg3* flies displayed climbing ability similar to the controls: all three genotypes exhibited a similar decrease in climbing ability with age (Figure 3D; age effect: *p* < 0.0001).

Spontaneous locomotor activity was assessed by counting the number of beam crossings per minute over 30 min, at the same time of the day as the other behaviors were tested. At that time of the day, *nlg3^Def1^* flies were not different from Cs when young; however, both *nlg3^Def1^* males and females (although not significantly) displayed increased activity when old (Figure 3E; *p* = 0.0032*;* 3F; *p* = 0.0806, respectively).

### 2.4. Aggression and dSO Avoidance Are Decreased in nlg3 Mutants in a Sex-Specific Manner

We were interested to know how the defects in social behavior were extended beyond social space. To accomplish this, we tested other social behaviors including sociability, aggression, and dSO avoidance. Experiments were performed as described previously for aggression [36], and dSO avoidance [37]; and adapted from earlier protocols in the case of sociability [11]. In sociability, no difference was observed in *nlg3^Def1^* flies compared to Cs at either age in males (Figure 4A; *p* = 0.9085) or females (Figure 4B; *p* = 0.3812).

There is no effect of age on aggression reported between 4–7 days old males [36], and we saw no significant effect of age either in our control males. Aggression in young *nlg3^Def1^* males was not different from that of the control; however older *nlg3^Def1^* males displayed reduced aggression (Figure 4C; *p* = 0.001). No difference in aggression was observed in females (Figure 4D; *p* = 0.3813).

Finally, the avoidance of the odorant left by stressed flies (dSO avoidance) was not different in young or old *nlg3^Def1^* males (Figure 4E; *p* = 0.1200), but *nlg3^Def1^* females had reduced avoidance at both ages (Figure 4F; *p* = 0.0128).

Taken together these results indicate that the effects of *nlg3* on social behavior are behavior- and sex-specific.

### 2.5. nlg3 Is Required for a Typical Response to the Social Environment

The social environment affects multiple behaviors in *Drosophila*, including social space. For example, social isolation leads to increased social space, i.e., reduced number of flies in proximity [14,15]. In addition, *nlg3* transcript abundance was shown to increase after social experience [34]. We wondered whether *nlg3* could be part of a pathway responsible for the environmental modulation of social space. Following single housing of Cs and *nlg3^Def1^* flies, we observed that both males Cs (as expected) and *nlg3^Def1^* flies had fewer flies within 4BL, although the decrease was not as pronounced in *nlg3^Def1^* flies (Figure 5A; effect of isolation: *p* < 0.0001; interaction between genotype and isolation: *p* = 0.0331). A similar result was observed in females (Figure 5B; isolation effect: *p* = 0.0006; Interaction: *p* = 0.0309).

We then determined the protein abundance of Nlg3 after isolation, in order to determine whether Nlg3 proteins levels vary in response to the environment. Western blot analysis of Cs fly head extracts revealed no difference in Nlg3 protein abundance after isolation (Figure 5C, D) in males (*p* = 0.9880) or females (*p* = 0.3415). Therefore, while the response to social space isolation is altered by the absence of *nlg3* expression, Nlg3 protein levels in control flies do not vary in response to social isolation. Further, we used group housed Cs males and females to determine if there was a sex difference in Nlg3 abundance. Females had lower abundance of Nlg3 than males (Appendix A; *p* = *0.001*).

## 3. Discussion

Here we report that the autism-related gene *nlg3* plays diverse roles in *Drosophila* behaviors. Social behaviors, including social space, aggression, and dSO avoidance were altered in *nlg3^Def1^* loss of function mutants in a sex-specific manner. Additionally, we confirm previously reported locomotor defects in young flies through startle-induced climbing and report spontaneous locomotion defects in older flies. Finally, we demonstrate that *nlg3* is required for a typical response to the environment in social space. All the results are summarized in Table 1.

We first provided evidence that only *nlg3^Def1^* had reduced protein abundance due to the deletion of *nlg3*. The mutants with P-element insertions, *nlg3^L04^* and *nlg3^GS32^*, had Nlg3 protein abundance similar to Cs. We expected *nlg3^L04^*, which incorporates a 3–4 kb P-element insertion, would be a truncated protein (Figure 1A). Incorporated into this insertion are stop codons along with splice acceptors in both the sense and antisense directions. The *nlg3^GS32^* fly line contains a 5.2 kb insertion into the regulatory region of *nlg3*. Due to such a large insertion, the disruption of gene expression would be expected [38]. However, it is possible that those insertions would affect *nlg3* regulation of transcription, through interfering with access to regulatory regions, modifying not the amount of product made, but how much and where it is made. In other words, Nlg3 protein could be expressed at different times (at earlier ages) or different sub-cellular locations (not the dendrite) in mutant flies. Such observations have been reported in mice models with the autism associated Nlg3 R451C knock-in. Those mice show no decrease in *nlg3* transcript abundance but reduced ability of protein processing, ER protein export and trafficking to the synapse [39,40,41]. Our study does not allow us to assess such aspects of Nlg3 protein dynamics, and *nlg3^L04^* and *nlg3^GS32^* needs further investigation.

As expected, *nlg3* mutant flies have a reduced climbing ability. Indeed, prior studies of *nlg3* mutants by Xing et al. [28] and Wu et al. [33] found that knockout flies have locomotor deficits. Both larval crawling and adult locomotion were significantly reduced [28]. Our analysis of mutant flies echoes this by showing *nlg3^Def1^* males and females have a significant reduction in climbing at the age of 3–4 days old, an age also tested by Xing et al. [28] and Wu et al. [33]. 

In contrast, at an older age we saw an increase in spontaneous activity and no effect on climbing ability. Xing et al. [28] and Wu et al. [33] did see decreased locomotion at 2–4 days old by monitoring locomotion over 24 h, or focusing at a time of the day with the most activity (ZT11–12) However, we monitored locomotion for 30 min, focusing at the time of day during which our social assays were performed (ZT4–8). As shown before [33], at that time of the day there is extremely limited spontaneous locomotion for both mutant *nlg3* and control genotypes. At ZT4–8, we only detected the deleterious effect of *nlg^Def1^* on locomotion when startling young flies. Additionally, overexpression of *nlg3* in males had a protective effect against the age-related decline in climbing ability. Finally, there was no correlation between locomotion and social space, as shown previously in several studies, including by Anderson, Scott and Dukas [42].

We next demonstrated that *nlg3* does play a role in social space, in both sexes, in an age-dependent manner. Male *nlg3^Def1^*, regardless of the age at which they were tested, have increased social space; however, female *nlg3^Def1^* were only further apart at an older age, revealing a genetic predisposition to abnormal social space as female flies age. Of note, we found that females Canton-S displayed lower levels of Nlg3, and that coincides with differences reported in transcripts levels in Flybase. Both *nlg3^L04^* and *nlg3^GS32^* flies were not different in social space. If these mutations are causing changes to spatial and temporal protein regulation, this could explain why we do not see large changes in social space. Mice with a R451C knock-in within the *nlg3* gene had normal sociability, but an increase in the number of interactions occurring [43]. This is not something investigated in our assay. Others have reported no effects of the R451C *nlg3* mutation on social interactions [44]. 

Although there was no significant effect of *nlg3^Def1^* on sociability, in other social behaviors the role of *nlg3* also appeared to be sex-specific. Male *nlg3^Def1^* flies were more aggressive at an older age, and females had decreased dSO avoidance at both ages.

This was the first time the role of *Drosophila nlg3* was examined in the context of social behavior. While the different assays can all be used to study social interactions between flies, we noted that the performances for each assay did not necessarily correlate with one another. Social space is resource independent, while other assays, like sociability, are not [11,14,17]. Differences in neural circuitry involved in the response to other individuals in each assay could also be responsible for the differences in social behavior.

The social spacing of females and males in response to variations in *nlg3* were different: males were farther apart, with either loss or overexpression of Nlg3, whereas females displayed a dose-effect: farther apart than the control with loss of *nlg3*, and closer with overexpression of *nlg3*. These sexually dimorphic responses in *nlg3^Def1^* flies suggest the presence of Nlg3 within, or synaptically in contact with, sexually dimorphic neurons, such as *fruitless*-expressing neurons [45].

As mentioned above, we also observed a role of *nlg3* in the response to the even moderate aging to which we subjected the flies, in a sex-specific manner. Only older *nlg3^Def1^* females displayed changes in social space, and only older *nlg3^Def1^* males were less aggressive. Finally, only the older male and female *nlg3^Def1^* flies exhibited elevated locomotion.

Lastly, despite no changes in Nlg3 protein levels in response to the social environment, we demonstrate that *nlg3* plays a role in that response. Social isolation has already been shown to affect courtship, courtship memory, and aggression [46,47,48,49], as well as neural excitability [49], chemical communication [50], sleep patterns [51], circadian rhythm [46], and olfactory memory [52]. Here we demonstrate a third type of social behavior affected by isolation, in which an increase in social space is observed, consistent with Simon et al. [14]. In *nlg3^Def1^* flies, the increase in social space observed in control flies was not as strong, providing evidence *nlg3* is involved in social space and is required for a typical response to the environment.

Nlg3 protein abundance did not change in control flies after isolation, nor at older ages (until 21 days—data not shown). This lack of change in Nlg3 protein contrasts with high-through studies reporting decreases in transcript levels, with social experience [34] and with age [53]. As indicated in Flybase, we also found an extremely low level of *nlg3* transcript expression (~1000-fold lower than the reference gene *ribosomal protein L32*, *rpl32*; Appendix A). The Nlg3 protein might be very stable, with either very low turnover, or its abundance (and possibly sub-cellular localization) might be tightly regulated at the translational level. Consequently, although *nlg3* is required for a response to the environment, other mechanisms or synaptic components sensitive to the environment are probably involved. We propose that either a neurotransmitter, or maybe an associated neurotransmitter receptor might be responding to the changes in social cues, either at the Nlg3 synapse or more probably upstream in the neurocircuitry.

The role of *nlg3* in *Drosophila* social behavior appears to be conserved, as *nlgs* are important for social behavior in worms, flies, mice, and humans [41,43,44,54]. Using *Drosophila’s* powerful genetic tools to understand how *nlg3* modulates social behavior and the response to the social environment will allow the identification of interacting genes and could lead to identifying new targets for pharmaceutical treatment of neuropsychiatric disorders such as autism.

## 4. Materials and Methods

### 4.1. Generation of nlg3 Deletion (nlg3^Def1^)

The *nlg3* deletion mutant was generated following FLP-mediated recombination between the FRT sites of the Exelixis pBAC insertion f00735 and P-element insertion XPd02461 as described [55]. DNA of single male progeny of female hs:flp; pBAC f00735 (3R: 7,648,768)/ XP d02461 (3R: 7,536,381) were screened by PCR using the following primer pairs: 

(1) Forward XP 5’-ACAACATCTTTGGTCATAAAATAGTCC-3’. Reverse (XP3”minus) 5’-TACTATTCCTTTCACTCGCACTTATTG-3’; 

(2) Forward (3pBacWH) 5’-CCTCGATATACAGACCGATAAAAC-3’. Reverse pBac 5’-ATAGTTTGCGTGCCTTTAGTTACC-3’.

Intra-chromosomal recombination results in a deletion between the two FRT sites in the insertions that were identified as double positive for PCR product for each primer pair. The deletion was validated using primers to the *nlg3* coding sequence and antibody staining.

### 4.2. Isolation of cDNA Containing nlg3 ORF and UAS Transgene for Overexpression

DNA fragments overlapping a unique BstZ17I restriction site were amplified by PCR from adult head cDNA of flies from a Canton-S strain carrying a mutation in the white *gene*. 

The following primer pairs were used: 

(1) A-F 5’-TAGGGCTTAGCACCTGCACT-3’ and A-R 5’-CTAAGCGCCTCCAGTGTTTC-3’; 

(2) B-F 5’-AACGAGATATTCGCCACGAT-3’ and B-R 5’-CATTTAAAACGCACGGTCAA.

Following DNA sequence confirmation, the fragments were joined at the BstZ17I site by ligation and subcloning. Asp718 (3’) and Not (5’) sites were added to the *nlg3* ORF fragment by PCR and sub-cloned into pUAST P-element transformation vector. P-element transformation and *Drosophila* transgenic strain generation was performed using standard procedures (Genetic Services, Inc).

### 4.3. Fly Stocks and Husbandry

All flies were reared in mixed sex groups inside bottles containing Jazzmix media (brown sugar, corn meal, yeast, agar, benzoic acid, methyl paraben, and propionic acid; Fisher Scientific, Whitby, ON, Canada) at 25 °C, 50% relative humidity on a 12:12 h light:dark cycle. Parents were a maximum of 14 days old, to limit variation in behavior observed with parents of older ages [20]. 

Five different fly lines were used in this study. *Drosophila melanogaster* Canton-S (Cs) was from our laboratory stock and used to outcross all lines used in this study 6 times, with the exception of *Elav*-*Gal4*. Three mutant *nlg3* genotypes of *Drosophila melanogaster* were used (see Figure 1A for a gene map): a line with a P-element insertion into the fourth intron (PBac{SAstopDsRed}LL04718, Kyoto stock center #140892), and a line with P-element insertion into the regulatory region of the gene (P{GSV1}neurG^S3205^ Kyoto stock center #205074). The deficiency line (*nlg3^Def1^*) and the insertion on the third chromosome of an upstream activation sequence (UAS) construct with the *nlg3* cDNA are described above. The pan-neuronal driver *embryonic lethal abnormal vision* (*elav*)-Gal4 was obtained from the Bloomington Stock Center (#876; P{w[+mC]=GAL4-*elav*.L}2/CyO).

### 4.4. Antibody Production

DNA corresponding to the complete cytoplasmic domain (AA 925-1148) of *nlg3* was amplified by PCR using forward primer 5’-GTGTACAACCAAAGGGACAAGACCCGAC-3’ and reverse primer 5’-CAAGCTTCACACGCAGCTCGTCCAT-3’, directionally cloned into the BsrG1 and HindIII sites of pET45b and expressed in *E. coli* using the Novagen pET system (EMD Biosciences, San Diago, CA, USA). Soluble His-tagged Nlg3-cyto protein was purified from bacterial extracts using Ni-Agarose chromatography following standard procedures, dialyzed against 1X PBS, and used as antigen to immunize guinea pigs (Open Biosystems/Thermo Fisher Scientific, Waltham, MA, USA).

### 4.5. Western Blot and Protein Analysis

Twenty male and female heads, separated by sex, were homogenized in 1X Laemmli sample buffer (32.9 mM Tris-HCl, 13% glycerol, 1% sodium dodecyl sulfate (SDS), 0.01% bromophenol blue) with 1% dithiothreitol (BioRad, Mississauga, ON, Canada). Protein lysates were separated on a 10% SDS-polyacrylamide gel (TGX FastCast Stain-Free gels; BioRad, Mississauga, ON, Canada) and electro-transferred to a nitrocellulose membrane. Proteins were incubated with a polyclonal guinea pig anti-Nlg3 antibody (1:4000) overnight at 4 °C, followed by horseradish peroxidase conjugated secondary antibodies (BioRad, Mississauga, ON, Canada; 1:10,000) and visualized using the ClarityMax Western enhanced chemiluminescence substrate (BioRad, Mississauga, ON, Canada). Western blots were analyzed with ImageLab software (BioRad, Mississauga, ON, Canada). All treatments were normalized to total protein by visualizing proteins on the membrane directly, following UV illumination of the blots [56,57] before enhanced chemiluminescence addition and detection occurred. All bands below the autofluorescence (probably resulting from the eye pigments) were included in the normalization. Full blot images including a molecular weight marker and total protein are shown in Appendix A. 

### 4.6. Fly Handling Prior to Behavior

All fly stocks were raised mixed sex in a socially rich environment (i.e., grouped-housed, with the exception of isolation conditions; see below). Apart for aggression experiments (see below under Aggression), newly eclosed flies from stock bottles were transferred to new bottles in order to age the flies. The day prior to a behavioral assay, 15–17 flies were collected and sexed under cold anesthesia. The morning of the experiment, all flies were transferred to new vials at least two hours prior to the assay and allowed to habituate to the test room conditions of 25 °C and 50% relative humidity. All experiments (apart for aggression) were performed under unified light and all replicates were tested in the same room between 12:00 p.m. and 4:00 p.m., corresponding to 4–8 ZT (Zeitgeber time: time after the onset of light), to reduce behavioral variation linked to diel periodicity. Sociability, aggression and activity monitor experiments were performed at McMaster, in the laboratory of one of the authors. The other experiments were all performed at Western University, in the other laboratory.

Flies were aged to either 3–4 (“young” flies) or 7–10 (“old” flies) days old for the experiments. These ages were chosen to examine the interaction that age has with genotype while avoiding age-related innate variations seen with the social space assay [22]. These ages also allow time for flies to mate to avoid social effects of mating status [14].

We always used an internal control for genetic background, as the entire data set display differences in performances, depending on the time of the year, and other variables that we cannot control (see for example Figure 2A–F).

### 4.7. Social Space Assay and ImageJ Analysis

Flies were placed into a triangular chamber and allowed to explore freely. Once flies settled in a stable group formation, pictures of each chamber were taken ~20–40 min after flies were placed in the chamber (i.e., time zero). Different ways of assessing social space have been used in past studies. In this study, we report how many flies were close to each focal fly. We chose to quantify the number of flies present within the distance of four body lengths of each fly, or ~1 cm, a metric also used by Xie et al. [15]. To acquire that information, images were processed using the free open access software ImageJ [58]. The new routines that we developed for these analyses on ImageJ are available at: https://github.com/flugrugger/bubble. 

Each replicate is the result of averaging the number of flies present within four body lengths for each fly in the chamber (12–17 flies/chamber, as there is no effect of variation in density on social space within that range [35]). Every data set consists of ~3 replicates per day and 3 independent days (9 replicates in total). Different testing days were separated by at least one week to control for environmental factors beyond our control.

### 4.8. Sociability

The sociability chamber consists of a circular arena (90 mm wide by 20 mm high) divided into 8 compartments with a hole in the center to allow flies to enter any compartment. Each compartment contains a patch of fresh food coated with a layer of grapefruit-yeast suspension (3 g yeast in 100 mL grapefruit juice) to enhance attractiveness. The chamber and performance of the assay is modified from Scott, Dworkin, and Dukas [11]. Sixteen same-sex flies enter the chamber by mouth aspiration through a hole in the lid and are allowed to roam freely for one hour. At this time, the number of flies in each chamber is counted and an aggregation index is calculated (sample variance divided by the mean number of flies in each chamber). The variance can take values between 0 and 32. For example, the least sociable option will have 8 chambers of 2 flies each, with a variance of 0. The most sociable situation will have 7 chambers of 0, and 1 chamber with all 16 flies. This will have a variance of 32, and therefore a maximum aggregation index of 16 (32/2). Twelve arenas per treatment were used.

### 4.9. Aggression

Flies were sexed within 6–8 h of eclosion and housed in isolation until 24 h before the test, because group-housed males are not very aggressive [59]. Approximately 24 h prior to testing, a non-focal fly of the opposite sex was aspirated into the focal fly’s vial and observed for mating. The non-focal flies were virgin 6-day-old flies of the same line as the focal fly they were paired with. After mating occurred, the non-focal flies were removed, and the focal flies were left in the vials until the test the following day. 

Flies were tested starting at 0 ZT, a time of high activity for the flies [60]. Two flies of the same genotype and sex were placed in polystyrene Petri dishes (35 mm diameter, 8 mm high) with a food patch in the center (5 mm diameter, 2 mm high). At the center of the food patch was a small (2 mm diameter) dot of yeast paste made from 1 part live yeast mixed with 2 parts grapefruit juice. All trials were video recorded for 20 min using Logitech HD Pro c920 webcams. Then, observers blind to fly age and genotype recorded the total duration of aggressive behaviors displayed by both flies in each arena using Behavioural Observation Research Interactive software (v7.9.7, Oliver Friard and Marco Gamba, Universita Delgi Studi Di Torino, Torino, Italy) [61]. Aggressive behaviors were defined by the ethogram outlined by Chen et al. [59], including occurrences of wing threat, lunging, high-level fencing, charging, holding, boxing and tussling. We tested 12 arenas (i.e., 24 flies) per line, per age, per sex for a total of 96 arenas across three replicates.

### 4.10. dSO Avoidance

The dSO avoidance assay was performed using a binary choice T-maze apparatus as described previously [12,37]. Fifteen responder flies of the same sex were added into the elevator of the Tmaze apparatus. Twenty flies of mixed sex (10 of each) were vortexed for 1 min (15 s on, 5 s rest, repeated 3 times) and subsequently removed from the vial, now containing dSO. This vial was placed on the Tmaze along with a fresh vial containing ambient air. Responder flies were allowed to choose between the air and dSO-containing vial for 1 min, followed by counting the number of flies in each vial. A performance index (PI; measure of the avoidance of dSO-containing vial) was calculated by subtracting the number of flies in the dSO-containing vial from those in the air vial and dividing by the total number of flies used in the assay. A higher PI indicates greater avoidance of dSO.

### 4.11. Climbing

Climbing was performed using the counter-current apparatus [62], as previously described [35]. In short, 40 flies separated by sex were mechanically banged to the bottom of the vial in the apparatus and allowed to climb to the top vial for 10 s. The number of flies reaching the top vial was counted and represented as the percent of the total flies used in the assay.

### 4.12. Activity

The activity assays were performed using two *Drosophila* activity monitors (Trikinetics Inc., software version 3.08, Waltham, MA, USA). The monitors were placed in an upright position such that the vials were held in the monitor in a horizontal position. Each monitor holds 32 vials (22 mm diameter, 48 mm long), and each vial is surrounded by infrared sensors that count the number of times a fly passes through them. We aspirated 1 fly per vial, and then placed the vials in the monitor (randomizing the position of different treatments within the monitor). The monitors were then placed inside opaque containers humidified to 55% relative humidity with a LED light from above. The flies were given 30 min to acclimate and then their activity was automatically recorded as the number of times each fly crossed the ring of infrared sensors that surrounded each vial each minute for a 30 min test period. We tested a total of 15 flies per line, per age, per sex for a total sample size of 120 flies across 3 replicates.

### 4.13. Social Isolation

To generate flies that were single housed, 2-day-old flies were collected and sorted into individual vials under cold anesthesia. Flies remained single housed for 7 days, similar to Simon et al. [14]. All flies used in social space and Western blot were grouped directly before being placed in the chamber for social space or homogenized in sample buffer for Western blot.

### 4.14. Statistical Analysis

We confirmed that the distributions of the data were analyzed following a Gaussian distribution prior to applying parametric tests and used an alpha level of 0.05 for all statistical tests. All analysis was completed using GraphPad Prism 8 (Prism version 8.3 for Mac, GraphPad Software, La Jolla California, USA, www.graphpad.com). All behavior used either one-way or two-way ANOVAs to test for the effects of genotype and age. When the data did not follow a Gaussian distribution (did not pass the normality tests), a Kruskal–Wallis test was used. Western blot analysis utilized a three-way ANOVA to test for effects of genotype, age, and protein isoform. *Post hoc* tests were performed to correct for multiple comparisons, after one-way ANOVA, and Kruskal–Wallis. Two-way ANOVAs and three-way ANOVAs by design test the effects of multiple factors (2 and 3, respectively) on an independent variable, as well as possible interactions between the variables. 

## Figures and Tables

**Figure 1 ijms-21-04601-f001:**
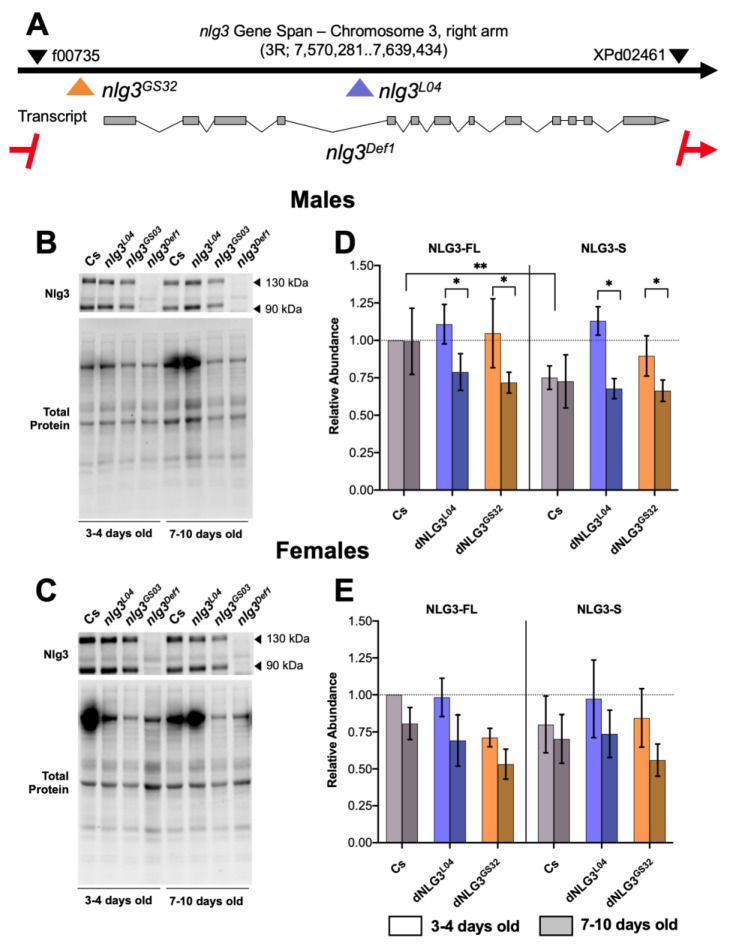
Map of neuroligin 3 (*nlg3*) mutations used in this study, and their effects on Nlg3 protein level. (**A**) Gene map of *nlg3* displaying insertion and deletion sites for corresponding mutants. The black arrow represents gene span. Grey squares in the transcript represent exons and lines indicate intronic regions. Colored arrows display insertion sites and the red line with arrows dashes represents the extent of the deletion of *nlg3* in the mutant *nlg3^Def1^*. All colors represent different genotypes throughout the data. Red: *nlg3^Def1^*; blue: *nlg3^L04^*; orange: *nlg3^GS32^*. Black arrows on the gene span represent the P-elements used to make the *nlg3^Def1^*. Insertion information can be found in the methods section. Information adapted from Flybase. (**B**,**C**) Representative Western blots for males (**B**) and females (**C**). Anti-Nlg3 immunoreactivity is displayed for Canton-S (Cs) and all mutants at 3–4 and 7–10 days old for both protein isoforms. (D,E) Mean protein abundance ± standard error of the mean (s.e.m.) in males (**D**) and females (**E**) for Cs, *nlg3^L04^,* and *nlg3^GS32^* flies. (**D**) There was no difference in protein abundance for Cs or *nlg3* mutants in males. Nlg3-FL was more abundant than Nlg3-S (three-way ANOVA—effect of protein isoform: *F_1,18_* = 9.196, *p* = 0.0072) and older flies had less Nlg3 protein than younger flies (three-way ANOVA—effect of age: *F_1,18_* = 5.236, *p* = 0.0344). (**E**) In females, there was no effect of genotype, protein isoform or age, although age was trending towards a decrease in protein abundance (*p* = 0.0727). All treatments are displayed as relative abundance to Nlg3-FL in Cs at 3–4 days old. Lighter colors represent 3–4 day-old flies. Darker colors represent 7–10 day-old flies. *n* = 4 for all treatments.

**Figure 2 ijms-21-04601-f002:**
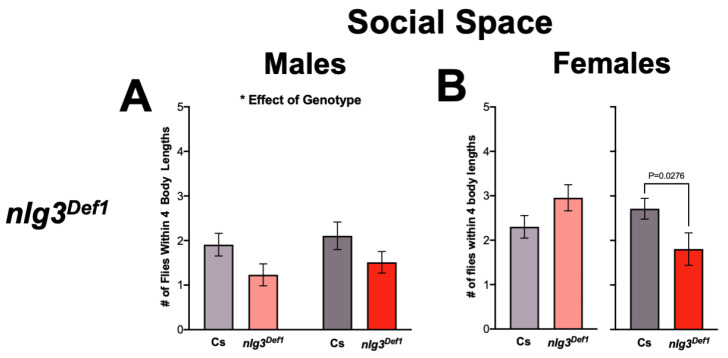
*Nlg3^Def1^* but not *nlg3^L04^* or *nlg3^GS32^* have altered social space. Mean number of flies within four body lengths (4BL) ± s.e.m. (**A**) Males *nlg3^Def1^* have significantly less flies within 4BL than male Cs, regardless of their age (two-way ANOVA: *F_1,49_*=5.538*, p* = 0.0227; 3–4 days old: *n* = 15 trials for Cs, *n* = 13 for *nlg3^Def1^*; 7–10 days old: *n* = 14 for Cs and *n* = 11 for *nlg3^Def1^*). (**B**) No difference was observed between young Cs and *nlg3^Def1^* females; however old *nlg3^Def1^* females had less flies within 4BL than Cs (Welch’s t-test: *T_13.62_*=2.097*, p* = 0.0276; 3–4 days old: *n* = 12 trials for Cs and *nlg3^Def1^*; 7–10 days old: *n* = 14 for Cs and *n* = 7 for *nlg3^Def1^*). The 7–10 day-old *nlg3^Def1^* and Cs females were tested on different days than 3–4 day-old flies, they were therefore plotted on separate axis and Welch’s t-tests were performed on the difference between *nlg3^Def1^* and Cs at each age. There was no effect of age, genotype, or any interaction for *nlg3^L04^* male (**C**) or female (**D**) and *nlg3^GS32^* male (**E**) or female (**F**). Color coding of bars corresponds to type of mutation: grey represents Cs, red represents *nlg3^Def1^*, blue represents *nlg3^L04^*, and orange represents *nlg3^GS32^*. The *p*-values were determined using a two-way ANOVA. Error bars represent s.e.m. (**C**,**D**): Males: 3–4 days old: *n* = 9 trials for Cs and *n* = 8 for *nlg3^GS32^*; 7–10 days old: *n* = 9 for Cs and *nlg3^GS32^*. Females: 3–4 days old: *n* = 9 for Cs and *n* = 8 for *nlg3^GS32^*; 7–10 days old: *n* = 9 for Cs and *nlg3^GS32^*. E-F: Males: 3–4 days old: *n* = 9 for Cs and *n* = 8 for *nlg3^L04^*; 7–10 days old: *n* = 9 trials for Cs and nlg3^L04^. Females: 3–4 days old: *n* = 9 for Cs and *n* = 7 for *nlg3^L04^*; 7–10 days old: *n* = 7 for Cs and *nlg3^L04^*. (**G**) The 3–4 day-old *elav > nlg3* males were not different from UAS-*nlg3*/+ control flies but *elav*-Gal4/+ flies had a higher number of flies within 4BL. *elav* > *nlg3* 7–10 day-old males were however more social than the two genotype controls (two-way ANOVA—effect of genotype: *F_2,30_* = 7.440*, p* = 0.0024*;* effect of age: *F_2,30_* = 0.02485*, p* = 0.8758*;* interaction of age and genotype: *F_1,30_* = 6.496*, p* = 0.0024; 3–4 days old: *n* = 6 trials for *elav*-Gal4/+, *n* = 6 UAS*-nlg3/+* and *n* = 5 for *elav* > *nlg3*; 7–10 days old: *n* = 6 for *elav*-Gal4/+, *n* = 8 for UAS*-nlg3*/+ and *n* = 6 for *elav* > *nlg3*). (**H**) Female *elav > nlg3* 3–4 days old had an increased number of flies within 4BL compared to *elav*-Gal4/+ and UAS*-nlg3*/+ controls. The 7–10 day-old *elav* > *nlg3* females had an intermediate phenotype with higher number of flies within 4BL than *elav*-Gal4/+ but lower number of flies within 4BL than UAS-*nlg3*/+ 7–10 days old (two-way ANOVA—effect of genotype: *F_2,32_* = 2.993, *p* = 0.0644; effect of age: *F_1,32_* = 9.240, *p* = 0.0047; interaction of age and genotype: *F_2,32_* = 3.633, *p* = 0.0379; 3–4 days old: *n* = 6 trials for *elav*-Gal4/+ and UAS-*nlg3*/+, *n* = 5 for *elav* > *nlg3*; 7–10 days old: *n* = 5 for *elav*-Gal4/+, *n* = 8 UAS-*nlg3*/+ and *elav* > *nlg3*). For (**A**–**H**): *n* = 5–15 trials–as detailed above–with 12–17 flies–always some escapees–per trial. Lighter colors represent 3–4 day-old flies. Darker colors represent 7–10 day-old flies.

**Figure 3 ijms-21-04601-f003:**
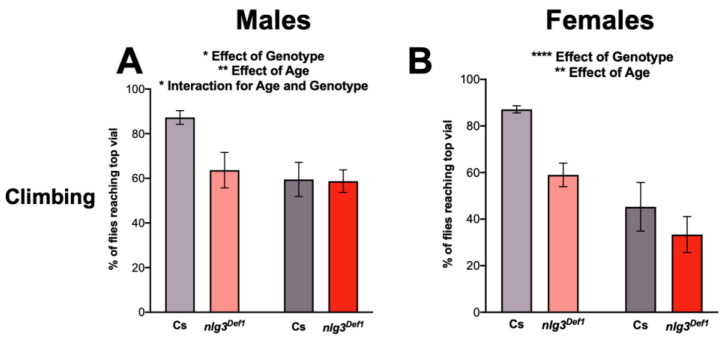
Locomotor activity is altered in *nlg3^Def1^* flies. (**A**,**B**) Startle-induced climbing represented by the mean percent ± s.e.m. of flies reaching the top vial. (**A**) *nlg3^Def1^* males had less flies than Cs reaching the top vial at 3–4 days old (two-way ANOVA –effect of genotype: *F_1,31_* = 4.464*, p* = 0.0428). However, this was not the case at 7–10 days old, where there was no difference between Cs and *nlg3^Def1^*. Cs males also had less flies reaching the top vial at 7–10 days old compared to 3–4 days old but this difference was not observed in *nlg3^Def1^* (two-way ANOVA—effect of age *F_1,31_* = 9.391*, p* = 0.0045; interaction between age and genotype *F_1,31_* = 4.780*, p* = 0.0365). (**B**) *nlg3^Def1^* females had less flies than Cs reaching the top vial at both ages (two-way ANOVA—effect of genotype: *F_1,30_* = 8.238*, p* = 0.0075); however flies at 7–10 days old had a lower percentage reaching the top vial compared to 3–4 day-old flies (two-way ANOVA—effect of age: *F_1,30_* = 23.38*, p* < 0.0001). *n* = 15 trials of 40 flies for all treatments. (**C**) *elav* > *nlg3* males had no difference in the number of flies reaching the top vial at 3–4 days old compared to the genotype controlled *elav*-Gal4/+ and UAS-*nlg3*/+ (two-way ANOVA—effect of genotype: *F_2,38_* = 1.809*, p* = *0.1776*). At 7–10 days old there was a decrease in the number of *elav*-Gal4/+ and UAS-*nlg3*/+ flies reaching the top vial (two-way ANOVA—effect of age: *F_1,38_* = 29.39*, p* < 0.0001). However, *elav* > *nlg3* males at 7–10 days old did not show this reduction in percentage of flies reaching the top vial (two-way ANOVA—interaction of age and genotype: *F_2,38_* = 3.993*, p* = 0.0267). (**D**) *elav* > *nlg3* females mirror this trend. The 3–4 day-old females also had no difference in the number of flies reaching the top compared to the genotype controls of the same age (two-way ANOVA—effect of genotype: *F_2, 36_* = 0.09254*, p* = 0.9118). As well, *elav*-Gal4*/*+ and UAS-*nlg3*/+ 7–10 day-old females decrease in the number of flies reaching the top vial (two-way ANOVA—effect of age: *F_1, 36_* = 42.12*, p* < 0.0001). The 7–10 day-old *elav* > *nlg3* females also had a reduction in the age-related decline of startle-induced climbing although not statistically significant (two-way ANOVA—interaction of age and genotype: *F_2,36_* = 1.760*, p* = 0.1865). **C**,**D**: *n* = 5–9 trials of 40 flies. (E,F) Spontaneous locomotor activity presented as violin plots of the mean number of beam crossings. (**E**) Age and genotype affect locomotor activity in males (Kruskal–Wallis: *p* = 0.0004), and male *nlg3^Def1^* had increased locomotor activity compared to Cs at 7–10 days old but not at 3–4 days old (Dunn’s *post hoc*: *p* = 0.0032). (**F**) In females a similar trend was observed (Kruskal–Wallis: *p* = 0.0367), where *nlg3^Def1^* 7–10 days old had higher activity than Cs (Dunn’s *post hoc*: *p* = 0.0806). **E**,**F**: *n* = 15 individual flies. Dotted horizontal lines represents 25–75% quartiles, solid horizontal line displays the median. Those are not visible when 75% of the data = 0. Lighter colors represent 3–4 day-old flies. Darker colors represent 7–10 day-old flies.

**Figure 4 ijms-21-04601-f004:**
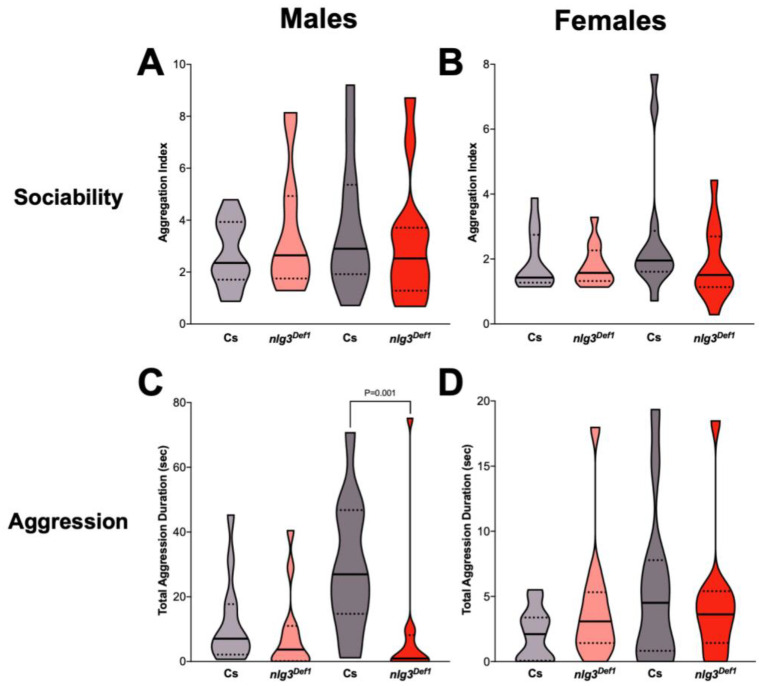
Aggression and *Drosophila* Stress Odorant (dSO) avoidance are altered in *nlg3^Def1^* flies in a sex-specific manner. (**A**,**B**) Violin plots displaying sociability index, for *n* = 12 trials of 16 flies. Sociability in males (**A**) was not different between *nlg3^Def1^* and Cs at either age (Kruskal–Wallis: *p* = 0.9085). The same effect was observed in females (**B**; Kruskal–Wallis: *p* = 0.3812). (**C**,**D**) Violin plots presenting total aggression duration, for 12 individuals for each treatment. Male (**C**) *nlg3^Def1^* and Cs were not different at 3–4 days old, but *nlg3^Def1^* old males were less aggressive than old Cs (Kruskal–Wallis: *p* = 0.0029, and Dunn’s *post hoc*: *p* = 0.001). Females were not different at either age (**D**; Kruskal–Wallis: *p* = 0.3813). **A**–**D**: Dotted horizontal lines represents 25–75 percent quartiles, solid horizontal line displays the median. (**E**,**F**) Mean performance index ± s.e.m. for the avoidance of dSO. No difference in PI was observed in males (3–4 days old: *n* = 13 trials for Cs, *n* = 15 for *nlg3^Def1^*; 7–10 days old: *n* = 12 for Cs and *nlg3^Def1^*). (**E**) However, *nlg3^Def1^* females had decreased PI compared to Cs at both ages (two-way ANOVA—effect of genotype: *F_1,43_* = 6.754*, p* = 0.0128; 3–4 days old: *n* = 14 trials for Cs, *n* = 13 for *nlg3^Def1^*; 7–10 days old: *n* = 12 for Cs and *nlg3^Def1^*). Lighter colors represent 3–4 day-old flies. Darker colors represent 7–10 day-old flies.

**Figure 5 ijms-21-04601-f005:**
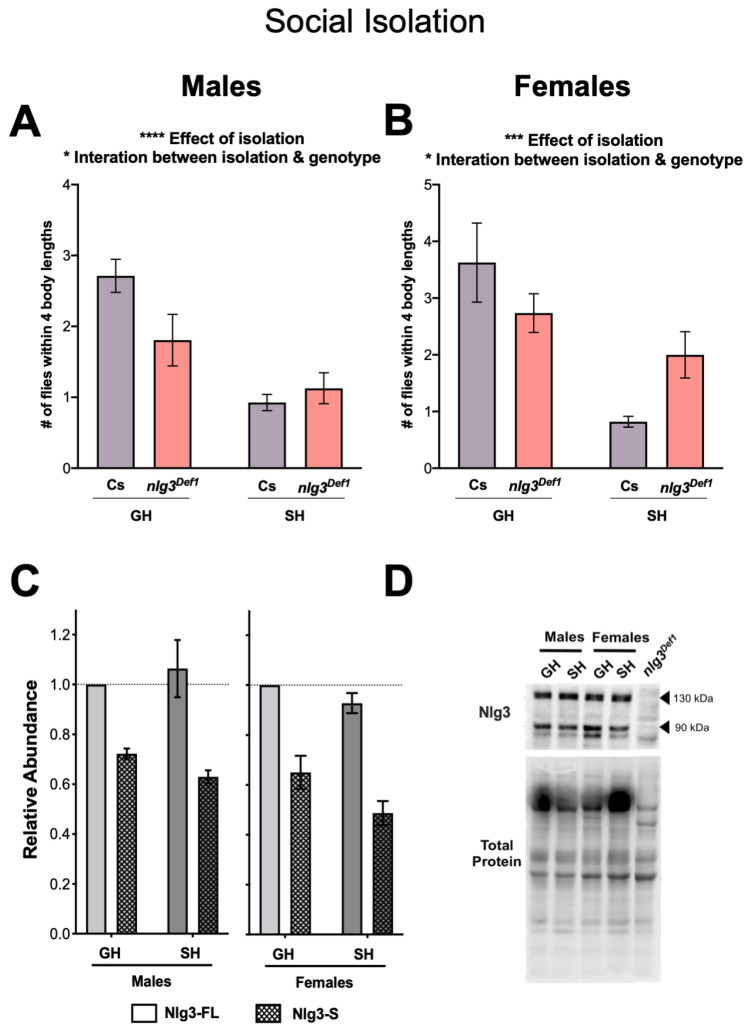
Social isolation: reduced response in social space in *nlg3^Def1^* flies, but no effect on protein levels in Canton-S. (**A**–**D**) Mean number of flies within 4BL ± s.e.m. in Cs, *nlg3^Def1^* and *nlg3^L04^*. *n* = 9 trials with 12–17 flies per trial. (**A**) Single housed Cs and *nlg3^Def1^* had a lower number of flies within 4BL after being single housed compared to group housed (two-way ANOVA—effect of isolation: F_1,32_ = 24.56, p < 0.0001); however SH *nlg3^Def1^* did not have as drastic a decrease compared to Cs (two-way ANOVA—interaction between isolation and genotype: F_1,32_ = 4.96, *p* = 0.0331). (**B**) Similar results as the males were obtained with *nlg3^Def1^* females (two-way ANOVA—effect of isolation: F_1,28_ = 15.10, *p* = 0.0006; two-way ANOVA—interaction between isolation and genotype: F_1,28_ = 5.166, *p* = 0.0309). (**C**) Nlg3 protein abundance in Cs males did not change in SH compared to GH flies in either protein isoform. The same Nlg3 expression pattern was observed in females. (**D**) Representative Western blots for Cs male and female GH or SH. Male treatments are relative to GH Nlg3-FL; males and female treatments are relative to GH Nlg3-FL females. GH: group housed, SH: single housed. All samples are normalized to total protein. *n* = 4 for all Western.

**Table 1 ijms-21-04601-t001:** Summary of behavior which performances significantly increased or decreased in *nlg^Def1^* males and females compared to sex and age-matched Cs.

			Males	Females
**Social Behaviours**	**Social Space**	3-4	🢆	ns
7-10	🢆	ns
Isolation	🢆	🢆
**Sociability**	3-4	ns	ns
7-10	ns	ns
**Aggression**	3-4	ns	ns
7-10	🢅	ns
**dSO Avoidance**	3-4	ns	🢆
7-10	ns	🢆
**Non-social Behaviours**	**Startle-Induced Climbing**	3-4	🢆	🢆
7-10	🢆	🢆
**Spontaneous Locomotor Activity**	3-4	ns	ns
7-10	🢆	ns
🢆 Indicates significant decrease compared to Cs		
🢅 Indicates significant increase compared to Cs		
ns: not significant

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
