# Peer review of "Abnormal Social Interactions in a Drosophila Mutant of an Autism Candidate Gene: Neuroligin 3"

_ijms, 2020, doi:10.3390/ijms21134601_

Round 1
Reviewer 1 Report
The manuscript by Yost et al., describes the impact of nlg3 gene on the social interactions and behavior of Drosophila melanogaster.
The manuscript is well-written, but some experiments need to be reconsidered and some other are needed to confirm the conclusions.
In my opinion, there are critical flaws in the Western blotting experiments that could affect the results described. In addition, some important details concerning the mutant lines are missing and some methods should be better described.
In conclusion, I cannot consider this manuscript acceptable until all these aspects will be clarified.
MAJOR POINTS
The western blot shown in figure 1 displays strikingly contrasting results compared to the expression data reported in Flybase (http://flybase.org/reports/FBgn0083963). How can it be possible that the authors could detect similar levels of proteins if the expression in females is barely detectable (if not zero)? How can authors reconcile with these data?
Furthermore, the proteolytic cleavage of Nlg3, as shown in the paper from Wu et al., 2017 produces a short product that is at least at a 1:10 ration if compared to the full length, whereas the two bands (130KD and 90 KD) shown in Figure 1 B and C are in a 1:1 ratio in the control. Considering that the same strain was used in this work and in the paper by Wu et al., it sounds strange that so dissimilar results were obtained here.
A confirmation of the "fuzzy" results obtained using this antibody is in figure 5 where an additional band appears just below the 90KD band detected in previous blots.
My final question is: are the authors sure what they are looking at? I fear that the antibody is giving misleading results. The original blot image, with the MWM, should be also provided as a supplementary file.
If the authors are sure of the antibody they have in their hands, then they should have validated it and this should be either shown or referenced.
How was the normalization made? Did the authors considered the whole lanes or just one or few bands within each lane?
The breakpoints of the deficiency used in this study should be reported in figure 1. Similarly, the genomic coordinates of the two insertions should be reported in the same figure.
Was the deletion in the def line characterized and validated? Again, this should either reported or referenced.
Revertants obtained from precise P-element excision should be tested to demonstrate the genotype-phenotypes link observed. In the absence of this experiment the conclusions drawn from the experiments cannot be claimed.
Minor points
page 2. ".....nlg2, nlg3, and nlg4 encode proteins located in the central nervous system (CNS) neurons and NMJ". In which developmental stage are these genes expressed?
page 3. "p-element" should be "P-element". Please change also other occurrence in the text
page 3. Why the Nlg3 protein is cleaved? This should be mentioned in the introduction since it has biological relevance.
Reviewer 2 Report
The study describes the impact of neuroligin 3 on the activity and social behavior of Drosophila melanogaster. Behavioral tests are complemented by quantification of Nlg3 protein in fly heads. Similar studies focusing on nlg1 and nlg4 have previously performed in Drosophila. Behaviours with different aspects of interactions (space, aggression, group formation etc.) are differently affected by nlg3 with respect to age and gender.
The present study does not find an increase of Nlg3 protein after isolation (although earlier studies reported an increase of nlg3 transcript levels) and does not reproduce effects of nlg3-deletion on locomotion, as reported in previous studies.
Since there is no emerging pattern concerning nlg3 gene function (without changes in Nlg3 protein levels!) with social vs non-social activity, young vs old flies and males vs females but only patchy effects appearing in some combinations of these factors, a prominent (or even causative) role of Nlg3 for the regulation of Drosophila social behaviors appears not to be likely.
General comments:
The authors should provide numbers of experiments and numbers of animals used in each experiment in the figure legends. All figure legends use normal letters (a, b, c,…) while in the figures capitals are used (A, B, C,…). Please adjust.
Please include medians in all violin plots.
Since the P-element insertions do not alter Nlg3 protein levels and don’t alter social spacing in flies, what is the value of showing the respective results in the respective figures?
Since the authors find opposite effects of nlg3 deletion on locomotion compared to previous studies, and report unchanged NLG3 protein levels after isolation (although earlier studies reported an increase of nlg3 transcript levels) they should also check nlg3 transcript levels in their experimental animals.
Abstract:
Line 4: “As previously reported ….”
Since the present study finds the opposite effect of nlg3-deletion on locomotion, this statement is misleading.
Introduction:
Page 2, L27-28: Move reference [29] directly after “ … clock neurons”, since it specifically refers to this statement.
Page 2, L38: Please add which tissue was analysed:.
“….transcript abundance in the head after a period of …..”
Results:
Page 3, L6: Please indicate (either here or in the introduction) what is known about full length and truncated isoform (function?, tissue distribution? etc.).
Page 3, L15: Replace “disappeared” by “was absent”. “Disappeared” suggests that the flies first had it (the difference) and subsequently lost it.
Page 5, L5: I cannot identify “blue line with arrows” in the figure.
Page 6, Figure 2: The difference between CS and respective mutant of younger males in 2A and 2E seems quite similar. One is significant but the other is not. Is that correct?
Labeling of the y-axis in Figure 2G is not correct.
Page 7, L10: “Reduced climbing was seen in ngl3-deleted females at both ages (Fig. 3B)”
Is this really true for older females, considering the significant overlap of error bars?
Figure 3: “Activity is altered in…”
Chose a more specific title for the figure.
Figure 3: Please adjust the y-axis in C and E to the extension (0-100%) used in A, B, D, F
Figure 3 E, F: Not obvious, why activity of young males is not different between CS and nlg3-deleted flies. If one would use a rank-based test, there would clearly be a difference.
Figure 3 E, F: I assume that the same numbers of CS and mutant flies were tested in the experiments. My understanding of a violin-plot would be, that the horizontal extension represents the numbers of flies with particular activity levels. Shouldn’t the area covered in each experimental group be similar, if similar numbers of animals were tested?
Figure 4F: Is the difference in younger females really significant?
Figure 5: “.. required for a typical response to the social environment …”. Be more specific towards what is shown in the figure.
Discussion:
Page 13, L4-5: “confirm previously reported locomotor defects through startle-induced and spontaneous locomotion”
Since the authors find the opposite effect of ngl3-deficiency as previous studies (increase instead of decrease), this statement is misleading.
Page 13, last paragraph: ”…possible that Nlg3 could be expressed at a different times or different sub-cellular locations in mutant flies”
Don’t understand the sense of this statement. Please clarify.
Page 14, L 8-10: Figure 1 in the study of Wu et al. 2018 shows that the locomotor activity of dnlg3 knockout flies is reduced over the entire 24 h period. “Picking the wrong period of testing” in the present study is therefore not a likely explanation of the opposite effect of dnlg3 deletion on spontaneous locomotion.
Page 14, L11-12: Please mention, HOW social space and locomotion was correlated in the two studies. Was it always in the same direction? This is a valuable information.
Page 14, L17: Both P-element insertions do NOT change overall Nlg3 protein content. My understanding is, that potential changes in social space must depend on some alteration. Why do you correlate unchanged behavior with unchanged “spatial/temporal protein regulation”?
Page 14, L19: What is the difference between “normal sociability” and “increased interactions”?
Page 14, L34: remove “that”.
Page 14, L36-38: I do not understand this point. Please explain.
Page 15, L4-9: This mainly refers to the data shown in Figure 5. Considering the complete loss of Nlg3 in the nlg3-deletion mutants, effects on “isolation-caused changes in interindividual distance” are quite minor and therefore do not justify the conclusion that Nlg3 plays a major role in this regulation. Moreover, Nlg3 protein levels are not affected by isolation, nevertheless behavioural changes occur. In addition there is this mismatch of isolation causing nlg3 transcript reduction (other studies) in contrast to unchanged Nlg3 protein levels (this study). The authors should quantify nlg3 transcript levels in flies from their colonies treated in their isolation-regimes to demonstrate equality (and comparability) with the earlier studies. Moreover, cDNA-treated (nlg3-overexpressing) flies should be compared between group-housed and isolated pretreatment, in order to correlate the behavioral observations with a “real” change in Nlg3 levels.
Materials and Methods
Page 16, L27-28: 12-17 flies per chamber were used in the experiments. For my understanding, it would be absolutely crucial to place exactly the same number of flies into an arena, if you want to analyse social space. Moreover, the description of this assay is not sufficiently detailed. The graphs displaying the results of these assays display “# of flies within 4 body lengths”. It is not clear to me what the values (usually between 2 and 5) mean.
Page 16, L 29: “… by at least one week to…”
Page 16: Sociability Assay
Based on the description, it is not clear how the authors come to a meaningful aggregation index. In the experiment 16 flies distribute among 8 sectors of the arena. The aggregation index results from the variance of the numbers of flies in each chamber divided by mean number of flies in each chamber. The mean number of flies in each chamber is always 2, no matter how the flies distribute among among the chambers. The variance can assume values between 0 and 14. So the expected values for the aggregation index should be between 0 and 7. However, the values extend these limits in graphs of the experimental results. Something must be wrong (or incomplete) with description.
Page 17, L2: remove “them”
Page 17, L8: “…the test on the following day”
Page 17, L9: Why are the aggression assays conducted at different times than the other behavioural assays?
Page 17, L 19-21: Not clear to me, which experimental animals were tested with which controls (neutral fighting partners?). You mention 1-day-old and 5-day-old bullies but the results shown in figure 4 include 3-4 and 7-10 day-old flies. Please clarify.
Page 17, L 21: “…Touch.”
Page 17. L38 following: In the text you call the climbing assay “startle-induced climbing assay”
Why? What is the “startle-component” in this assay?
Page 18, L 18-25: The authors perform multiple comparisons on several occasions of their statistical analysis, which requires corrections to avoid false positives. However, nothing like this is mentioned in the description of statistical procedures.
Author Response
Please see attachement

Round 2
Reviewer 2 Report
The authors provided reasonable responses to my comments on the first version of the manuscript. The manuscript was considerably improved. However, there remain some important issues that should be solved before I can recommend publication.
1) The present study does not reproduce effects of nlg3-deletion on locomotion, as reported in previous. This point was already mentioned in the first report and the authors disagreed with my point of view. However, I totally disagree with their view on this point.
Xing et al. 2014 and Wu et al. 2018 used the same setup to monitor spontaneous locomotor activity of adult Drosophila. Both studies (measuring activity for different periods of time) found a reduction of spontaneous locomotion of dnlg3-deleted flies in the range of ~50%. This study using the same experimental apparatus, in contrast, finds unchanged locomotor activity in young flies and (though not significantly) increased activity in old mutant flies which is a behavior change in the opposite direction compared to the previous studies.
So the authors cannot claim in their abstract and elsewhere in the manuscript:
“As previously reported, locomotion is affected by a loss of function nlg3 mutation”
The authors further try to justify their point by referring to their results from startle-induced climbing assays, which indeed show a reduced number of mutant flies, that reach a certain height in the test apparatus. However, “startle-induced climbing” is different from “spontaneous horizontal locomotion”. Flies may actively walk but still not direct to the top , just because their gravity-sensing is somewhat impaired. Observing decreased performance of mutants in the climbing assay, together with no (or even opposite) changes in spontaneous locomotion does not justify to say, that the study found the same effects on locomotion as previous studies.
2) As requested, the authors included medians and 25/75 percentiles in their violin plots. Especially in the older animals, these are hardly distinguishable from the pattern of the violin. Please chose a more obvious presentation.
3) Supplemental Figure 3: Please indicate briefly, what rpl32 codes for, since it is not mentioned in the materials section.
4) Although the authors incist that “labeling of the Y-axis in Fig. 2G is correct”, it is not. The Y-axis has the numbers “0 – 1 – 1 – 2 – 2 – 3” which doesn’t make sense.
5) Figure 3 is entitled “Locomotor activity is altered in ….” but it also includes data from climbing assays. As mentioned above, I regard locomotion and climbing as different behaviors. Animals may actively run around in the climbing assay but may not reach the top, just because their gravity-sensing is impaired. So locomotion (= walking distance) is different from climbing (= reaching a certain height with orientation against gravity). This difference should be reflected in the entire manuscript.
6) My comment to first version:
Figure 3 E, F: I assume that the same numbers of CS and mutant flies were tested in the experiments. My understanding of a violin-plot would be, that the horizontal extension represents the numbers of flies with particular activity levels. Shouldn’t the area covered in each experimental group be similar, if similar numbers of animals were tested?
Authors’response: The number of beam crossings are independent on the number of flies used in each assay. The violin plots are displaying the distribution of the data, not the number of animals.
Comment to the response:
I do not understand this explanation. Activity of each fly in the assay was determined for each minute of the test period. So each fly tested should provide the same number of data points for average activity (= number of beam crossings) per minute. If the same numbers of Cs and mutant flies are tested, the total numbers of data points should be identical and hence the area covered by the violin plot should be identical. In case that many flies how 0 beam crossings in many 1-minute-intervalls, the extension of the horizontal line at “0” should be large (and contribute to the overall area covered by the respective violin). So the data do not represent the real distributions of flies’ activity, in the way they are presented.
7) Old commentary: 26. Page 14, L34: remove “that”.
Author’s response: We are not sure what is the reviewers referring to. If it is the “that” is the following sentence: ” These sexually dimorphic responses in nlg3Def1 flies suggest that the presence of Nlg3”, we should not remove it – it is grammatically correct as is, as confirmed by 7 of the co-authors, who are native English speakers.
Comment to the response: Please read the entire sentence. If you leave “that” in, the sentence is not complete!
8) Old comment: Page 16, L27-28: 12-17 flies per chamber were used in the experiments. For my understanding, it would be absolutely crucial to place exactly the same number of flies into an arena, if you want to analyse social space. Moreover, the description of this assay is not sufficiently detailed. The graphs displaying the results of these assays display “# of flies within 4 body lengths”. It is not clear to me what the values (usually between 2 and 5) mean.
Authors’ response:
We now can expand on the description of social space, and explain that density between 12 and 17 does not alter social space, as shown by McNeil et al. 92015).
“Social space assay and ImageJ analysis Flies were placed into a triangular chamber and allowed to explore freely. Once flies settled in a stable group formation, pictures of each chamber were taken ~20-40 minutes after flies were placed in the chamber (ie. time zero). Different ways of assessing social space have been used in past studies. In this study, we report how many flies were close to each focal fly. We chose to quantify the number of flies present within the distance of 4 body length of each fly, or ~ 1 cm, a metric also used by Xie et al. [15]. To acquire that information, images were processed using the free open access software ImageJ [55]. The new routines that we developed for these analyses on ImageJ are available at: https://github.com/flugrugger/bubble. Each replicate is the result of averaging the number of flies present within 4 body lengths for each fly in the chamber (12-17 flies/chamber – as there is no effect of variation in density on social space within that range [33]). Every data set consists of ~3 replicates per day and 3 independent days (9 replicates total). Different testing days were separated by at least one week to control for environmental factors, beyond our control. “
New comment:
Reference [33] is not correct for the respective statement. I guess that [35] was meant. In that study by McNeil et al., different parameters were used to measure social distance in fly groups. It is not clear to me, whether one can claim from their results that variable group size of 12-17 flies has no influence on interindividual distance. However, if would plan to study spacing of flies in an arena, I would most importantly keep the numbers of flies identical, which is a very simple goal to be achieved. I still think that this is a very crucial point which cannot be neglected in the analysis.
9) Page 9, first lines:
“… decrease in climbing ability”
The flies’ ability to climb was not tested, only their “climbing activity”.
10) Page 16, line 3: “locomotor climbing ability”
This is everything mixed!
